 **eLIFE**

# Identification of polarized macrophage subsets in zebrafish

**Mai Nguyen-Chi[1,2†], Béryl Laplace-Builhe[1,2†], Jana Travnickova[2,3], Patricia Luz-Crawford[1,2], Gautier Tejedor[1,2], Quang Tien Phan[3], Isabelle Duroux-Richard[1,2], Jean-Pierre Levraud[4,5], Karima Kissa[2,3], Georges Lutfalla[2,3], Christian Jorgensen[1,2,6‡], Farida Djouad[1,2*‡]**

[1]Institut de Médecine Régénérative et Biothérapies, Institut national de la santé et de la recherche médicale, Montpellier, France; [2]Université de Montpellier, Montpellier, France; [3]Dynamique des Interactions Membranaires Normales et Pathologiques, Centre national de la recherche scientifique, Montpellier, France; [4]Macrophages et Développement de l'Immunité, Institut Pasteur, Paris, France; [5]Département de Biologie du Développement et Cellules Souches, Institut Pasteur, Paris, France; [6]Clinical unit for osteoarticular diseases and Department for Biotherapy, Centre Hospitalier Universitaire, Montpellier, France

**\*For correspondence:** farida. djouad@inserm.fr

†These authors contributed equally to this work

‡These authors also contributed equally to this work

**Competing interests:** The authors declare that no competing interests exist.

**Abstract** While the mammalian macrophage phenotypes have been intensively studied in vitro, the dynamic of their phenotypic polarization has never been investigated in live vertebrates. We used the zebrafish as a live model to identify and trail macrophage subtypes. We generated a transgenic line whose macrophages expressing *tumour necrosis factor alpha* (*tnfa*), a key feature of classically activated (M1) macrophages, express fluorescent proteins *Tg(mpeg1:mCherryF/tnfa: eGFP-F)*. Using 4D-confocal microscopy, we showed that both aseptic wounding and *Escherichia coli* inoculation triggered macrophage recruitment, some of which started to express *tnfa*. RT-qPCR on Fluorescence Activated Cell Sorting (FACS)-sorted tnfa[+] and tnfa[−] macrophages showed that they, respectively, expressed M1 and alternatively activated (M2) mammalian markers. Fate tracing of *tnfa*[+] macrophages during the time-course of inflammation demonstrated that pro-inflammatory macrophages converted into M2-like phenotype during the resolution step. Our results reveal the diversity and plasticity of zebrafish macrophage subsets and underline the similarities with mammalian macrophages proposing a new system to study macrophage functional dynamic.

## Introduction

Behind the generic name 'macrophage' hides various cell types with distinct phenotypes and functions. Currently, it is well established that macrophages are not just important immune effector cells but also cells with critical homeostatic roles, exerting a *myriad* of functions in development, homeostasis, and tissue repair and playing a pivotal role in disease progression (*Wynn et al., 2013*). Therefore, there is a high interest in a better characterization of these cells to establish an early and accurate diagnosis. The wide variety of macrophage functions might be explained by the outstanding plasticity and versatility of macrophages that efficiently respond to environmental challenges and changes in tissue physiology by modifying their phenotype (*Mosser and Edwards, 2008*). Although there is a consensus that macrophages are a diversified set of cells, macrophage subtypes are still poorly characterized. Indeed, although these cell populations have been extensively investigated in mouse and human, these studies were mostly performed in vitro using monocyte-derived macrophages induced under specific stimuli. A comprehensive characterization of macrophage

**eLife digest** Inflammation plays an important role in helping the body to heal wounds and fight off certain diseases. Immune cells called macrophages—which are perhaps best known for their ability to engulf and digest microbes and cell debris—help to control inflammation. In mammals, different types of macrophage exist; the most functionally extreme of which are the M1 macrophages that stimulate inflammation and M2 macrophages that reduce the inflammatory response.

Macrophages acquire different abilities through a process called polarization, which is controlled by signals produced by a macrophage's environment. Polarization has been well investigated in human and mouse cells grown in the laboratory, but less is understood about how this process occurs in live animals.

Nguyen Chi, Laplace-Builhe et al. investigated whether zebrafish larvae (which are naturally transparent) could form an experimental model in which to investigate macrophage polarization in living animals. Zebrafish were first genetically engineered to produce two fluorescent proteins: one that marks macrophages and one that marks M1 macrophages. These fluorescent proteins allow the movement and polarization of macrophages to be tracked in real time in living larvae using a technique called confocal microscopy. Nguyen Chi, Laplace-Builhe et al. also isolated macrophage cells from these zebrafish at different times during the inflammatory process to identify which macrophage subtypes form and when.

The results show that unpolarized macrophages move to the sites of inflammation (caused by wounds or bacterial infection), where they become polarized into M1 cells. Over time, these M1 macrophages progressively convert into an M2-like macrophage subtype, presumably to help clear up the inflammation.

Furthermore, Nguyen Chi, Laplace-Builhe et al. show that the M1 and M2 macrophage subtypes in zebrafish are similar to those found in mammals. Therefore, genetically engineered zebrafish larvae are likely to prove useful for studying macrophage activity and polarization in living animals.

subsets that takes into account their specific behaviour, phenotypic diversity, functions, and modulation shall rely on a real-time tracking in the whole organism in response to environmental challenges.

Mouse and human macrophages have been classified according to their polarization state. In this classification, M1 macrophages, also referred as classically activated macrophages, are pro-inflammatory cells associated with the first phases of inflammation, while M2 macrophages, also known as alternatively activated macrophages, are involved in the resolution of inflammation and tissue remodelling (*Gordon, 2003*; *Biswas and Mantovani, 2010*; *Sica and Mantovani, 2012*). Differential cytokine and chemokine production and receptor expression define the polarization state of macrophages. However, it is worthwhile to note that such binary naming does not fully reflect the diversity of macrophage phenotypes in complex in vivo environments in which several cytokines and growth factors are released and adjust the final differentiated state (*Chazaud, 2013*; *Thomas and Mattila, 2014*). Macrophages might adopt *intermediate activation phenotypes* classified by the relative levels of macrophage subset-specific markers. Therefore, macrophage plasticity results in a full spectrum of macrophage subsets with a myriad of functions (*Mosser and Edwards, 2008*; *Xue et al., 2014*). Although the possible phenotype conversion of macrophages from M1 to M2 has been suggested in in vitro studies, a recent study argues for the sequential homing of M1 and M2 macrophages to the site of injury (*Stout et al., 2005*; *Sica and Mantovani, 2012*; *Shechter et al., 2013*). Such controversies highlight the lack of accurate real-time tracing of macrophage subtypes in vivo in the entire animal.

Inflammation is a model of choice to study the wide range of macrophage subsets involved from its initiation to its resolution. Therefore, in the present study, we propose to decipher in vivo in real time the kinetic of macrophage subset recruitment, their behaviour and their phenotypic plasticity at the molecular level during a multiple-step inflammatory process. We used the zebrafish larvae model for its easy genetic manipulation, transparency, and availability of fluorescent reporter lines to track macrophages (*Ellett et al., 2011*). While the existence of macrophage subtypes in zebrafish embryos

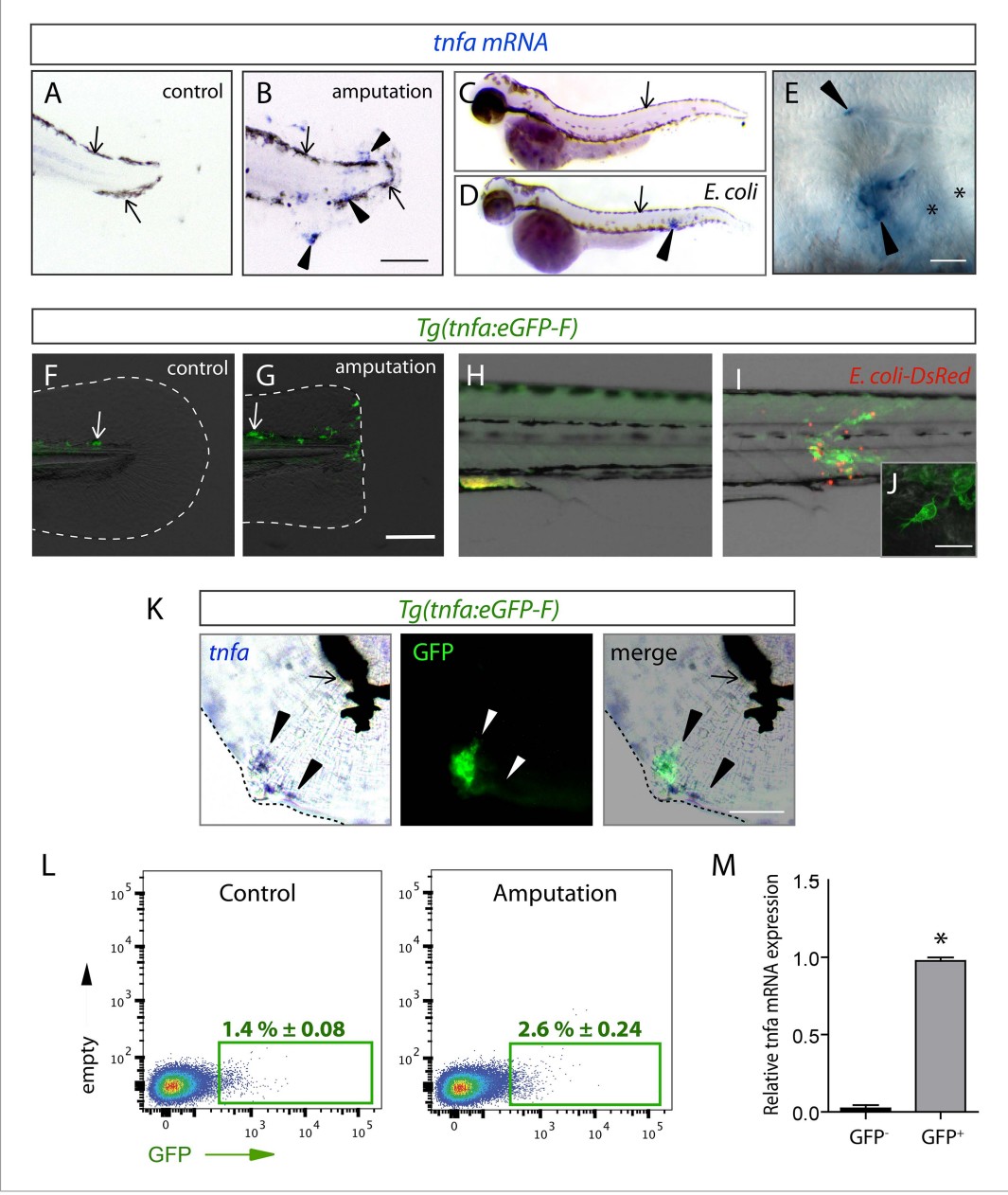

**Figure 1.** The *(tnfa:eGFP-F)* reporter line recapitulates transcriptional activation of *tnfa* upon wound-induced inflammation and *Escherichia coli* infection. (**A–E**) *Tumour necrosis factor alpha* (*tnfa*) mRNA expression (blue, arrowhead) was detected by in situ hybridization using *tnfa* anti-sense probe: at 6 hpA in (**A**) intact (control) and (**B**) amputated fins from 3 dpf WT larvae, (**C**) in uninfected larvae (54 hpf, hours post-fertilization) and (**D, E**) *E. coli* infected larvae (24 hpi, 54 hpf). Arrows show melanocytes (black). (**E**) Imaging of *tnfa* mRNA expression in the muscle at higher magnification, asterisks show muscle fibres, scale bar in (**B**) = 100 µm and in (**E**) = 50 µm. (**F, G**) eGFP fluorescence (green) was analyzed by fluorescent microscopy in (**F**) intact (control) and (**G**) amputated *Tg(tnfa:eGFP-F)* fins at 6 hpA, dotted lines outline the caudal fin, scale bar = 100 µm and at 16 hpi in *Tg(tnfa:eGFP-F)* larvae injected with (**H**) PBS or (**I, J**) *E. coli* (red) in the muscle. Arrows show auto-fluorescent xanthophores. (**J**) Multi-scan confocal analysis of GFP expression in *E. coli*-infected *Tg(tnfa:eGFP-F)* larvae, scale bar = 20 µm. (**K**) *tnfa* mRNA and eGFP-F expressions were analyzed using microscopy at 6 hpA in amputated fins from 3 dpf *Tg(tnfa:eGFP-F)* larvae. Dotted lines delimit the caudal fin, arrowheads show overlapping signals, and arrows show the pigments. Scale bar = 100 µm. (**L**) Graphed data of representative fluorescence-activated flow cytometry analysis of eGFP+ cells in upon amputation. *Tg(tnfa:eGFP-F)* larvae were either kept intact (control) or amputated at 3 dpf, and cells were collected at 6 hr post-treatment. Green gates represent eGFP+ population and mean percentage of eGFP+

*Figure 1. continued on next page*

*Figure 1. Continued*

population ±s.e.m is indicated. (**M**) Relative expression of *tnfa* in eGFP- and GFP⁺ cells in amputated larvae. Real-time RT-PCR on separated cells using EF1a as a reference gene. Graph represents the mean value of three independent experiments ±s.e.m. *p < 0.05.

has been suggested, they have not been fully characterized (**Herbomel et al., 1999**; **Ellett et al., 2011**; **Cambier et al., 2013**; **Petrie et al., 2014**). Here, we report a new reporter transgene for TNFa, a central inflammatory cytokine and well-established marker of M1 macrophages, instrumental to discriminate macrophage subsets during intravital imaging.

## Results and discussion

### In vivo visualization of macrophage activation and polarization

Fin wounding-induced inflammation and *Escherichia coli* inoculation in zebrafish larvae of 3 dpf are two well-established models triggering macrophage recruitment. Using in situ hybridization, we observed that the expression of the *tumour necrosis factor alpha* (*tnfa*), a consensus marker of M1 macrophages, was induced in cells accumulated in the caudal fin and the muscle following amputation ($n_{larvae} = 29/33$) and *E. coli* inoculation ($n_{larvae} = 12/12$), respectively (**Figure 1A–E**). To study the cells that express the *tnfa* transcripts, we established the *Tg(tnfa:eGFP-F)* transgenic zebrafish line expressing a farnesylated (membrane-bound) eGFP (eGFP-F) under the control of the *tnfa* promoter. While eGFP-F was undetectable in intact fins of *Tg(tnfa:eGFP-F)* larvae ($n_{larvae} = 10/10$), it was expressed in cells recruited to the wound at 6 hr post-amputation (hpA) ($n_{larvae} = 16/16$, **Figure 1F,G**). Similarly, eGFP-F expression was upregulated in cells accumulated in the muscle of *Tg(tnfa:eGFP-F)* larvae at 16 hr post-inoculation (hpi) of DsRed-expressing *E. coli* ($n_{larvae} = 8/8$), compared to Phosphate-buffered saline (PBS) injection ($n_{larvae} = 3/3$, **Figure 1H,I**). Confocal analysis confirmed the presence of a membrane-bound eGFP in cells displaying a typical myeloid morphology (**Figure 1J**). To demonstrate that the *Tg(tnfa:eGFP-F)* line recapitulates transcriptional activation of *tnfa*, we performed a simultaneous detection of *tnfa* mRNA by in situ hybridization and GFP-F protein by immunofluorescence in amputated larvae 6 hpA. We observed a consistent overlap between *tnfa* and GFP-F signal in the fin ($n_{larvae} = 11/11$), showing the direct correlation of eGFP-F and *tnfa* transcriptional activation in the fin of the reporter line (**Figure 1K**). In addition, we FACS-sorted GFP⁺ cells from wounded *Tg(TNFa:eGFP-F)* larvae 6 hpA and performed RT-qPCR to analyze *tnfa* expression. We observed a significant increase of *tnfa* mRNA level in eGFP⁺ cells as compared to eGFP⁻ cells (**Figure 1L,M**). All together these results indicate that the *Tg(TNFa:eGFP-F)* reporter line recapitulates transcriptional activation of *tnfa*. Then, with the ability to specifically track *tnfa*-expressing cells, we used *Tg(tnfa:eGFP-F)* fish to study macrophage activation by mating them with *Tg(mpeg1:mCherryF)* fish in which macrophages express farnesylated mCherry (mCherryF) under the control of the macrophage-specific *mpeg1* promoter (**Ellett et al., 2011**; **Nguyen-Chi et al., 2014**). In intact *Tg(tnfa:eGFP-F/mpeg1:mCherryF)* larvae, no eGFP-F was observed in macrophages (**Figure 2A**). We imaged double transgenic larvae *Tg(tnfa:eGFP-F/mpeg1:mCherryF)* using 4D confocal microscopy from 45 min post-amputation and found that macrophages were recruited to the wound from 1 hpA, some starting to express eGFP from 3 hpA (**Figure 2B,C** and **Video 1**). From 5 hpA already activated macrophages, that is, expressing *tnfa* arrived at the wound (**Video 1** and **Figure 2C**). Similarly, infection with a crimson-expressing *E. coli* in the muscle induced the expression of *tnfa* in phagocytes few hours following the infection (**Figure 2—figure supplement 1**, **Video 2**). Imaging of the double transgenic larvae *Tg(tnfa: eGFP-F/mpeg1:mCherryF)* showed that *tnfa*-expressing phagocytes were mainly macrophages (**Figure 2—figure supplement 1**, **Video 2**). These results show the dynamic macrophage activation in real-time in vivo including recruitment and rapid phenotypic change. During the revision of this paper, a similar result has been published (**Sanderson et al., 2015**).

### Morphology and behaviour of macrophage phenotypes

To test whether *tnfa*⁺ and *tnfa*⁻ macrophages harboured different cellular characteristics, we first analyzed their morphology in fin-wounded *Tg(tnfa:eGFP-F/mpeg1:mCherryF)* larvae. *tnfa*⁺*mpeg1*⁺ cells displayed flattened and lobulated morphology (**Figure 2D**), while *tnfa*⁻*mpeg1*⁺ were elongated

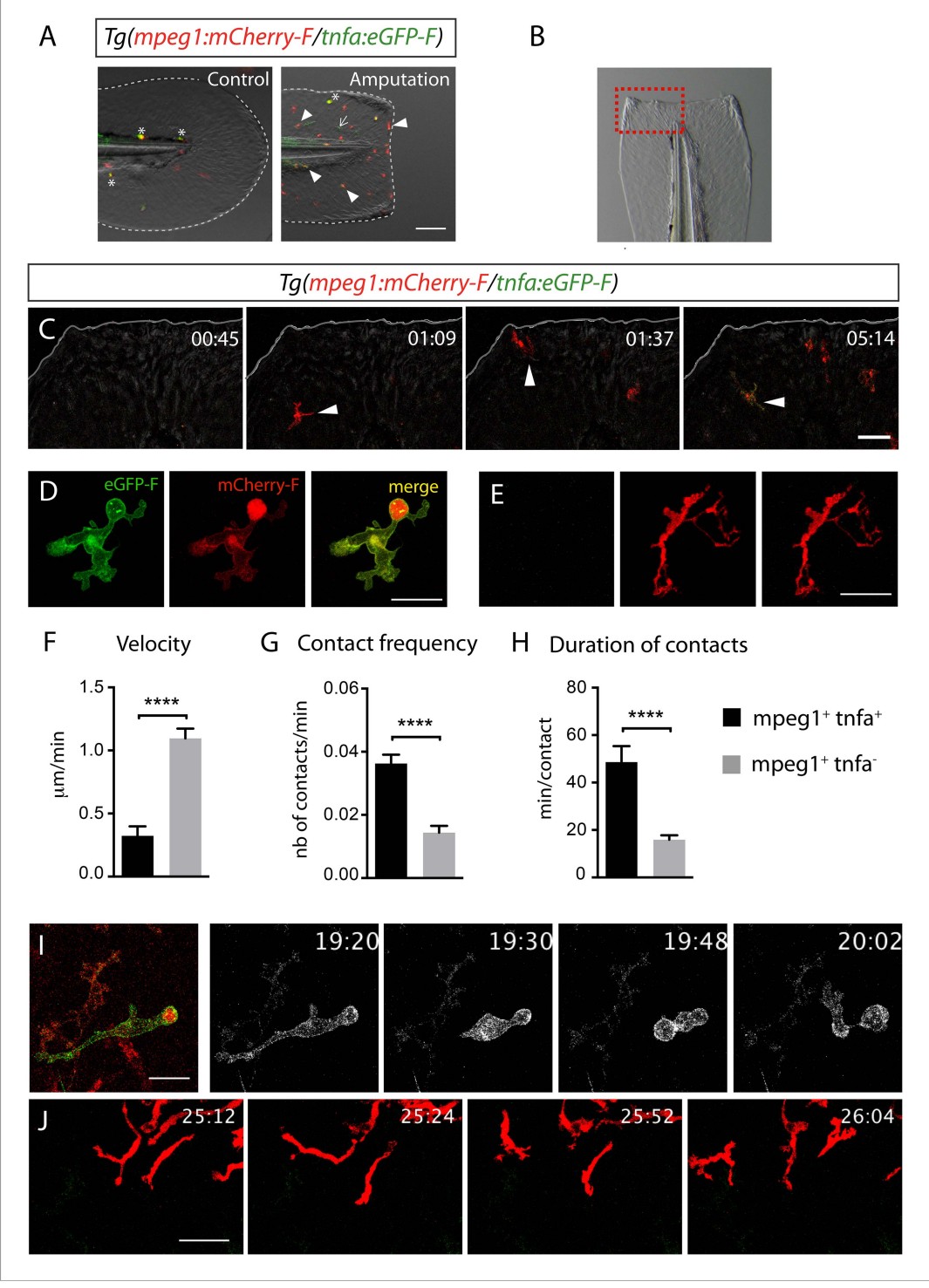

**Figure 2**. Activation, morphology, and behaviour of TNF-α⁺ macrophages in *(tnfa:eGFP-F/mpeg1:mCherry-F)* transgenic larvae upon wound-induced inflammation. (**A**) eGFP-F (green) and mCherryF (red) fluorescence was analyzed by fluorescent microscopy in intact (control) and amputated *Tg(mpeg1:mCherryF/tnfa:eGFP-F)* fins at 6 hpA of 3 dpf larvae. Arrowheads show recruited macrophages that express *tnfa*, arrows show tnfa⁺ cells that are not macrophages, and asterisks show auto-fluorescent pigments. Dotted lines outline the caudal fin, scale bar = 100 μm. (**B**) Bright-field image of the wounded fin of a 3 dpf *Tg(mpeg1:mCherryF/tnfa:eGFP-F)* larva. Dotted red box shows the region imaged in **C**. (**C**) Representative time-lapse maximum projections show the activation of macrophages arriving at the wound in 3 dpf amputated *Tg(mpeg1:mCherryF/tnfa:eGFP-F)*. The time pA is shown on top right
*Figure 2. continued on next page*

*Figure 2. Continued*

corner and indicated in hours and minutes, white lines outline the caudal fin. The transcriptional activation of *tnfa* (green) in recruited macrophage (red, arrowhead) was first observed from 3 hpA. Scale bar = 30 μm. White lines outline the caudal fin. (**D**, **E**) Maximum projections of confocal analysis of eGFP-F (green) and mCherryF (red) expressions in recruited macrophages at (**D**) 18 hpA and (**E**) 24 hpA in *Tg(mpeg1:mCherryF/tnfa:eGFP-F)*. *tnfa⁺ mpeg1⁺* macrophages exhibit a round and protrusive morphology, while *tnfa⁻mpeg1⁺* macrophages exhibit a dendritic morphology. (**F**) Velocity of *tnfa⁺mpeg1⁺* and *tnfa⁻mpeg1⁺* macrophages (N = 18). (**G**) Frequency of macrophage–macrophage contacts and (**H**) time length of the contacts of *tnfa⁻mpeg1⁺* and *tnfa⁺mpeg1⁺* cells. Measurements were extracted from three independent videos of amputated *Tg(mpeg1:mCherryF/tnfa:eGFP-F)*, for contact frequency, N = 15 and for duration of the interaction, N = 11 macrophages. ****p < 0.0001.
(**I**) Representative time-lapse maximum projections show the behaviour of *tnfa⁺mpeg1⁺* macrophages, starting 19h20 pA during 42 min. Two macrophages (green + red) interact by cell–cell contact. These macrophages (eGFP in grey) remain attached up to 40 min. Scale bar 20 = μm. (**J**) Representative time-lapse maximum projections show the behaviour of *tnfa⁻mpeg1⁺* macrophages, starting 25h12 pA during 52 min. Macrophages (red) barely establish cell–cell contact. Scale bar = 30 μm.

The following figure supplement is available for figure 2:

**Figure supplement 1**. Activation of tnfa⁺ macrophages in *(tnfa:eGFP-F/mpeg1:mCherry-F)* transgenic larvae upon *E. coli* infection.

and dendritic (*Figure 2E*). As we observed that *tnfa⁺mpeg1⁺* cells were predominant at the wound at 18 hpA and *tnfa⁻mpeg1⁺* cells at 24 hpA (data not shown), we imaged the behaviour of these macrophage populations in wounded fins from *Tg(tnfa:eGFP-F/mpeg1:mCherryF)* larvae at these time points. *tnfa⁺mpeg1⁺* cells presented a lower velocity (0.32 μm/min) than *tnfa⁻mpeg1⁺* macrophages (1.09 μm/min, *Figure 2F*) but a higher cell–cell contact frequency (0.036 VS 0.016 contacts/min) with other macrophages (*Figure 2G,I,J* and *Videos 3, 4*). Measurements of the duration of macrophage–macrophages contacts showed that these contacts lasted longer (48.6 min/contact) than that of *tnfa⁻mpeg1⁺* macrophages (15.9 min/contact, *Figure 2H–J* and *Videos 3, 4*). All together these data highlight different morphology and behaviour of macrophage phenotypes in live zebrafish suggesting the existence of macrophage subsets exhibiting different functions.

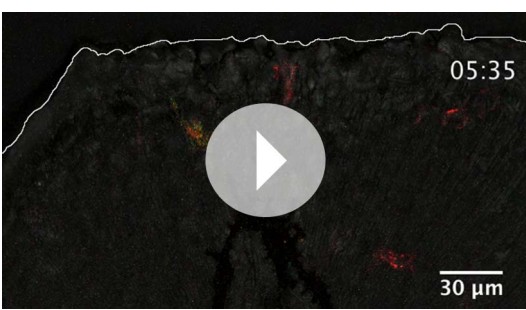

**Video 1.** Transcriptional activation of *tnfa* in macrophages of *(tnfa:eGFP-F/mpeg1:mCherry-F)* transgenic larvae upon amputation. Representative time-lapse maximum projections show the transcriptional activation of *tumour necrosis factor alpha* (*tnfa*) in macrophages arriving at the wound in 3 dpf amputated *Tg(mpeg1:mCherryF/tnfa:eGFP-F)*. The time pA is shown on top right corner, white line outline the caudal fin. Scale bar = 30 μm. Image stacks were acquired every 3 min 30 s from 45 min pA to 7 hr 48 min pA at 2-μm intervals, 1024 × 512 pixel resolution using a confocal microscope TCSSP5 SP5 inverted equipped with a HCXPL APO 40×/ 1.25–0.75 oil objective (Leica). Excitation wavelengths used were 488 nm for EGFP-F and 570 nm for mCherryF.

## Macrophages phenotypes are activated in a time-dependant manner

To quantify the respective frequency of *tnfa⁺* macrophages (mCherry⁺eGFP⁺ referred as dbl⁺) and *tnfa⁻* macrophages (mCherry⁺eGFP⁻ referred as mCh⁺), we performed flow cytometry analysis on cells isolated from *Tg(tnfa:eGFP-F/ mpeg1:mCherryF)* larvae at different time points following caudal fin amputation or *E. coli* inoculation (*Figure 3A,B*). While only 5.6% ± 0.9 (s.e.m.) *dbl⁺* cells were detected in the *mpeg1⁺* population of the intact larvae, a steady increase of the *dbl⁺* population from 6 to 20 hpA (up to 27.33 ± 0.2%) was observed. This percentage decreased dramatically at 26 hpA to 8.75 ± 1%. In *E. coli* inoculation experiments, the frequency of *dbl⁺* cells increased as soon as 3 hpi (55.60 ± 0.6%) and remained stable until 26 hpi. These results demonstrate that wound-induced macrophage activation is transient compared to infection-induced macrophage activation.

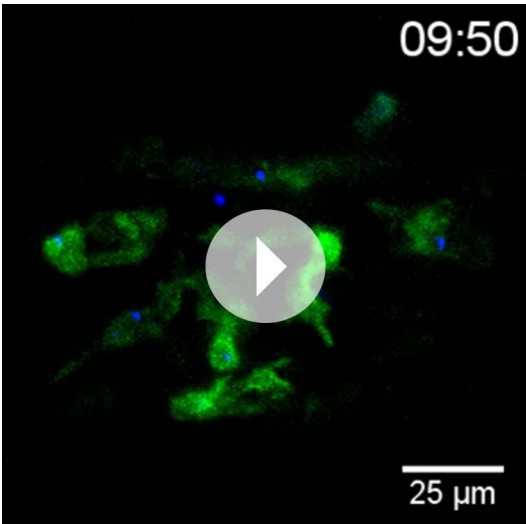
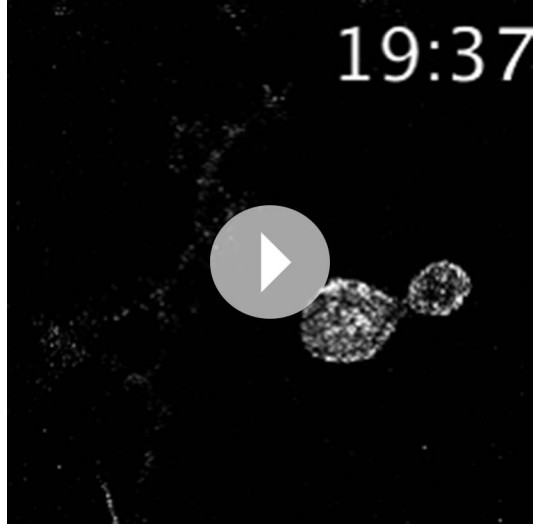

**Video 2.** Transcriptional activation of *tnfa* of *(tnfa: eGFP-F/mpeg1:mCherry-F)* transgenic larvae upon *E. coli* infection. *Tg(mpeg1:mCherryF/tnfa:eGFP-F)* larvae were infected with crimson-expressing *E. coli* (blue) at 3 dpf in the muscle and imaged from 30 min pi to 10 hr 30 min pi. Representative time-lapse maximum projections show the expression of *tnfa* (green) induced in myeloid-like cells at the infection site from 3 hpA. The time pA is shown on top right corner, scale bar = 25 μm. Image stacks were acquired every acquired 3 min 30 s at 2-μm intervals, 512 × 512 pixel resolution with a X2 zoom using a confocal microscope TCSSP5 SP5 inverted equipped with a APO 20× objective (Leica). Excitation wavelengths used were 488 nm for EGFP-F and 580 nm for Crimson. To distinguish Crimson from mCherry, emission filter was adjusted from 630 to 750 nm.

**Video 3.** *tnfa*[+] macrophage behaviour following amputation. Representative time-lapse maximum projections show the behaviour of *tnfa*[+]*mpeg1*[+] macrophages in *Tg(mpeg1:mCherryF/tnfa:eGFP-F)* larvae following amputation, starting 19h20 pA during 42 min. Two macrophages *tnfa*[+]*mpeg1*[+] interact by cell–cell contact. These macrophages (GFP in grey) remain attached up to 40 min. Scale bar = 20 μm. Image stacks were acquired every 3 min 30 s at 2 μm-intervals, 1024 × 512 pixel resolution using a confocal microscope TCSSP5 SP5 inverted equipped with a HCXPL APO 40×/1.25–0.75 oil objective (Leica). Excitation wavelengths used were 488 nm for EGFP-F and 570 nm for mCherryF.

## Molecular signature of tnfa[+] and tnfa[−] macrophage populations

To characterize at the molecular level *tnfa*[−] and *tnfa*[+] macrophage populations during early and late phases of inflammation, we FACS-sorted dbl[+] and mCh[+] cells from *Tg(tnfa:eGFP-F/mpeg1:mCherryF)* tail-amputated larvae (*Figure 3C*) and analyzed them by qRT-PCR. In mammals, M1 and M2 macrophages are reported to be involved, respectively, in the initial phase of inflammation and in the resolution phase. Cell sorting was thus performed at 6 hpA and 26 hpA following caudal fin amputation since the kinetic analysis of macrophage subset activation (*Figure 3B*) suggested that these two time points correspond to initiation and resolution of inflammation, respectively. As expected, high levels of *mpeg1* expression was observed in the mCh[+] and dbl[+] sorted cells at 6 and 26 hpA (*Figure 3D*), and high levels of *tnfa* expression was detected in double-positive populations at 6 hpA (*Figure 3E*). These observations demonstrated that fluorescence of these transgenes can be efficiently used to track and separate macrophage sub-populations. At 6 hpA, dbl[+] macrophages expressed high levels of *tnfb*, *il1b*, and *il6* compared to mCh[+] macrophages (*Figure 3F,G*) that are well-known markers of M1 macrophages in mammals (*Mantovani et al., 2002*; *Martinez et al., 2006*). By contrast, mCh[+] macrophages expressed low levels of these pro-inflammatory cytokines at both 6 and 26 hpA (*Figure 3F,G*), but expressed high levels of *tgfb1*, *ccr2*, and *cxcr4b* (*Figure 3H*), that are specifically expressed in mammalian M2 macrophages (*Mantovani et al., 2002*; *Martinez et al., 2006*; *Hao et al., 2012*; *Beider et al., 2014*; *Machado et al., 2014*). Of note, neither *Arginase 1* (*Arg1*), which is largely used as a M2 marker in mouse but not in human (*Chinetti-Gbaguidi and Staels, 2011*; *Pourcet and Pineda-Torra, 2013*), nor *il10* (data not shown), a known M2 marker in mammals (*Mantovani et al., 2002*), was detected in zebrafish macrophages. Importantly, based on

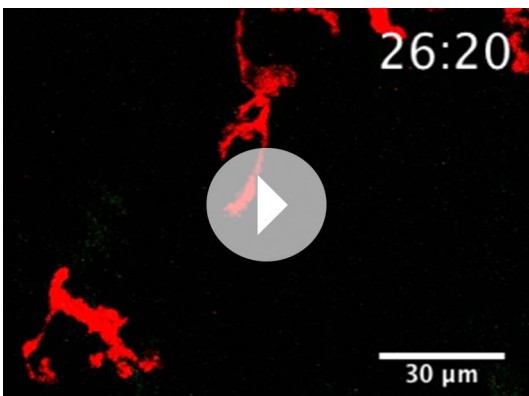

**Video 4.** *tnfa⁻* macrophage behaviour following amputation. *Tg(mpeg1:mCherryF/tnfa:eGFP-F)* caudal fins were amputated at 3 dpf. Representative time-lapse maximum projections show the behaviour of *tnfa⁻ mpeg1⁺* macrophages, starting 25h12 pA during 42 min. Macrophages (red) scarcely establish cell–cell contact. Scale bar = 30 µm. Image stacks were acquired every 4 min at 2 µm-intervals at 1024 × 512 pixel resolution using a confocal microscope TCSSP5 SP5 inverted equipped with a HCXPL APO 40x/1.25–0.75 oil objective (Leica). Excitation wavelengths used were 488 nm for EGFP-F and 570 nm for mCherryF.

the stability of the eGFP in *Tg(tnfa:eGFP-F/mpeg1:mCherryF)* larvae allowing us to specifically track the behaviour and fate of pro-inflammatory macrophages, we found that the dbl⁺ pro-inflammatory macrophages changed their phenotype at 26 hpA. Indeed, dbl⁺ macrophages negative for M2 markers at 6 hpA, displayed at 26 hpA, in parallel to a significant decrease of *tnfa*, *il1b*, *and il6* expression level, a significant increased expression level of *ccr2* and *cxcr4b* (*Figure 3E,H*). Of note, a tendency toward differential expression level was observed for *tnfb* and *tgfb1* between 6 and 26 hpA. To go further and demonstrate that the same macrophages are present at the wound site during inflammation and its resolution, we generated the *Tg(mpeg1:GAL4/UAS:Kaede)* larvae to track macrophages exploiting the conversion of the native green fluorescence of Kaede into red fluorescence under UV light. Recruited macrophages were photoconverted 6 hpA and imaged at 26 hpA revealing that early recruited macrophages were still present at the wound area 20 hr later (*Figure 4—figure supplement 1*). Then, GFP⁺ macrophages were specifically tracked using time-lapse imaging of wounded *Tg (mpeg1:mCherryF/TNFa:GFP-F)* fins from 6 to 26 hpA. We show that initially recruited eGFP⁺ macrophages remain at the injury site and still express the GFP (*Figure 4A–C* and *Video 5*). The analysis of macrophage behaviour over time shows that among eGFP⁺ macrophages displaying an amyboid phenotype at the wound edge 6 hpA, 50% change toward a fibroblastic phenotype from 11 hpA when they moved distally (*Video 5*). All together these data show that pro-inflammatory macrophages underwent a phenotypic conversion toward an intermediate phenotype in which both M1 and M2 markers are expressed. In addition, this molecular characterization of macrophages in zebrafish reveals the conservation of macrophage subtypes between zebrafish and human.

In conclusion, we identified macrophage subsets in zebrafish and described their behaviour and fate during a process of inflammation (*Figure 4D*). Live imaging of transparent transgenic zebrafish larvae allowed the first real-time visualization of macrophage activation and polarization. In parallel, a molecular analysis of macrophage sub-populations highlights the evolutionary conservation of macrophages from fish to mammals. We propose that in response to wounding zebrafish, unpolarized macrophages are recruited to the inflammation site and adopt a M1-like phenotype. Subsequently, they progressively convert their functional phenotype from M1-like to M2-like in response to progressive inflammatory microenvironment changes within the tissue (*Figure 4D*). Live imaging of the new transgenic line we generated opens new avenues to study in real time in live vertebrates the full spectrum of macrophage activation, polarization, and functions.

## Materials and methods

### Ethics statement

All animal experiments described in the present study were conducted at the University of Montpellier according to European Union guidelines for handling of laboratory animals (http://ec.europa.eu/environment/chemicals/lab_animals/home_en.htm) and were approved by the Direction Sanitaire et Vétérinaire de l'Hérault and Comité d'Ethique pour l'Expérimentation Animale under reference CEEA-LR-13007.

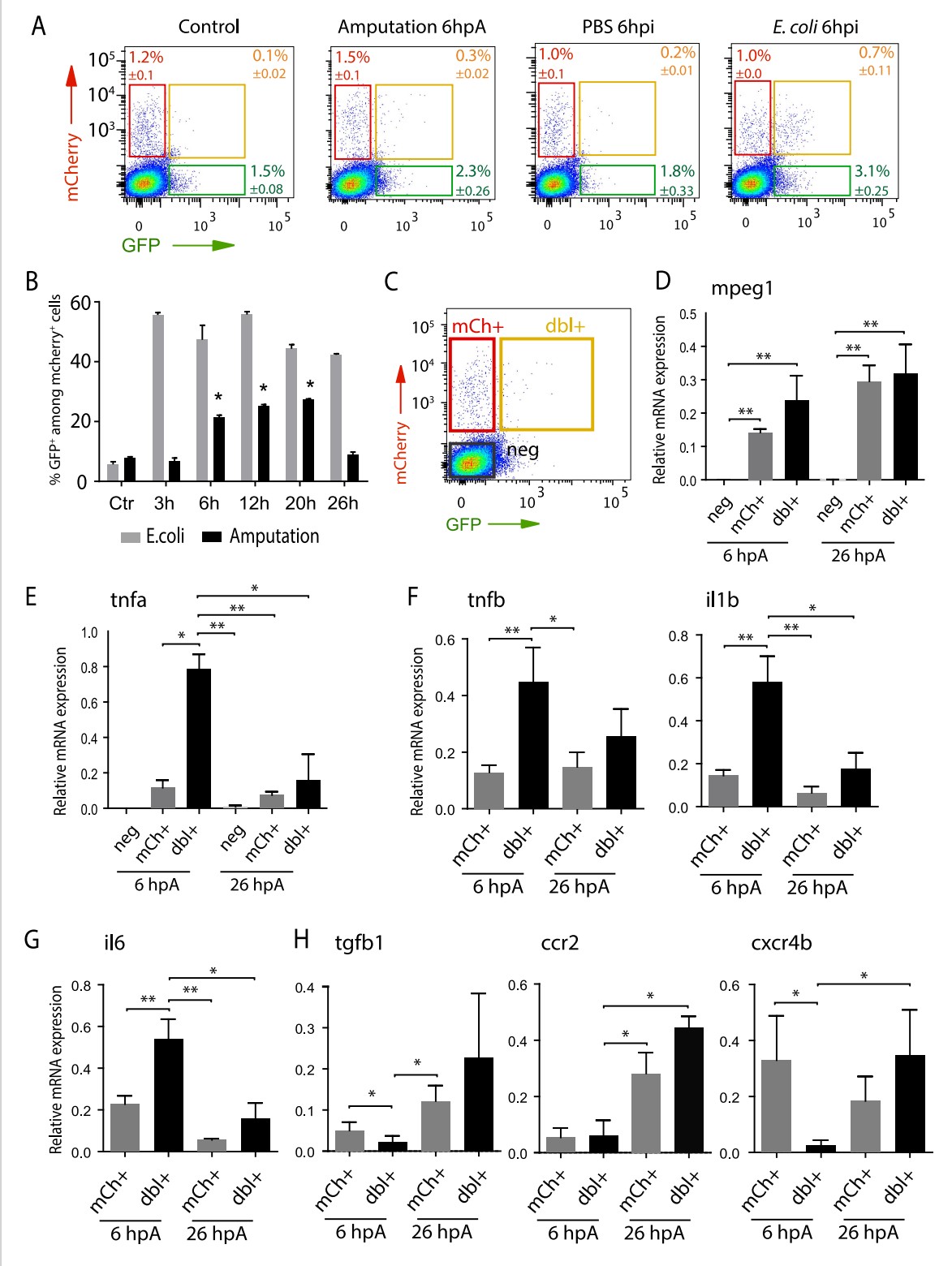

**Figure 3.** Isolation and molecular characterization of macrophage phenotypes. (**A**) Graphed data of representative fluorescence-activated flow cytometry analysis of *tnfa*+ and *tnfa*− macrophages upon inflammatory stimulations. *Tg(mpeg1:mCherryF/tnfa:eGFP-F)* larvae were either kept intact (control), or amputated, or injected with PBS or with *E. coli* at 3 dpf, and cells were collected at 6 hr post-treatment. Red, green, and yellow gates represent mCherry+, eGFP+, and mCherry+eGFP+ populations, respectively. (**B**) Graph represents the kinetic of the frequency of *mpeg1*+*tnfa*+ macrophages in macrophage population (*mpeg1*+) in three independent experiments following stimulation: amputation and *E. coli* infection (*E. coli*) at indicated time points. *p < 0.05
*Figure 3. continued on next page*

*Figure 3. Continued*

vs 3 hpA, mean value of three experiments ±s.e.m. (**C**) Gating strategy to isolate control cells (mCherry⁻ eGFP⁻, neg), *tnfa*⁻ macrophages (mCherry⁺ eGFP⁻, mCh⁺), *tnfa*⁺ macrophages (mCherry⁺eGFP⁺, dbl⁺). (**D–H**) Relative expression of (**D**) *mpeg1*, (**E**) *tnfa*, (**F**) *tnfb*, *il1b*, (**G**) *il6*, (**H**) *tgfb1*, *ccr2*, and *cxcr4b* in cells neg, mCh⁺, and dbl⁺. *Tg(mpeg1*:mCherryF/*tnfa:eGFP-F)* were amputated at 3 dpf and cells were collected and separated at 6 hpA and 26 hpA. Real-time RT-PCR on separated cells using *EF1a* as a reference gene. Graph represents the mean value of five independent experiments ±s.e.m. Statistical significance between bars are indicated *p < 0.05, **p < 0.01.

## Zebrafish line and maintenance

Fish and embryo maintenance, staging, and husbandry were as previously described (*Nguyen-Chi et al., 2014*). Experiments were performed using the AB zebrafish strain (ZIRC) and the transgenic line *Tg(mpeg1:mCherryF)* to visualize macrophages. For the photoconversion experiments, a cross of Tg (*mpeg1:Gal4)^gl25* and Tg(*UAS:kaede)^rk8* lines was used, using breeders selected for progeny with negligible silencing of the UAS transgene.

## Transgenic line construction

The TNFa promoter (Gene ID: 405785) was amplified from zebrafish genomic DNA using primers zTNFaP4 (CCCGCATGCTCCACGTCTCC) and zTNFaE11N (TTATAGCGGCCGCCCGACTCTCAAGCTTCA). The resulting fragment was phosphorylated using T4PNK, digested by NotI and cloned in a farnesylated eGFP (eGFP-F) derivative of pBSKI2 (*Thermes et al., 2002*). The resulting plasmid (pI2promTNFa: eGFP-F) harbours a 3.8-kb fragment of the zebrafish *tnfa* promoter, including part of the first coding exon. It uses the endogenous ATG codon of *tnfa* to drive the translation of eGFP-F. The expressed eGFP-F harbours the first 7 amino acids of zebrafish TNFa at its N-terminus (MKLESRA). The expression cassette is flanked by two I-SceI sites. pI2promTNFa:eGFP-F was co-injected in fertilized eggs with the enzyme I-SceI (New England Biolabs, France). Developing embryos were injected with non-pathogenic *E. coli* at 3 dpf (days post-fertilization), and those that developed a specific green fluorescence were raised as putative founders. The offspring of putative founders was tested the same way in order to establish the stable transgenic line. Low expression of eGFP in the pharynx at 3 dpf was used to check larvae for the presence of the transgene.

## Larva manipulation for inflammation assays and imaging

Caudal fin amputation was performed on 3 dpf larvae as described in *Pase et al. (2012)*. The caudal fin was transected with a sterile scalpel, posterior to muscle and notochord under anaesthesia with 0.016% Tricaine (ethyl 3-aminobenzoate, Sigma Aldrich, France) in zebrafish water. Larvae were inoculated at 3 dpf by $2.10^3$ CFU *E. coli* K12 bacteria harbouring either DsRed (*van der Sar et al., 2003*) or Crimson (Clontech, France) expression plasmid or no plasmid. Imaging was performed as previously described (*Nguyen-Chi et al., 2014*) using a confocal TCS SP5 inverted microscope with a HCXPL APO 40×/1.25–0.75 oil objective (Leica, France). Image stacks for time-lapse videos were acquired every 3–5 min, typically spanning 30–60 µm at 2-µm intervals, 1024 × 512 or 512 × 512 pixel resolution. The 4D files generated from time-lapse acquisitions were processed using Image J. They were compressed into maximum intensity projections and cropped. Brightness, contrast, and colour levels were adjusted for maximal visibility. Velocity of macrophages was measured using Manual Tracking Image J plugin. Frequency of macrophage–macrophage interaction and duration of interactions were measured manually on stack images. For tracking of macrophages, eGFP-F⁺ mCherryF⁺ cells from *Tg(mpeg1*:mCherryF/*tnfa:eGFP-F)* wounded fins were tracked using time-lapse image series from 6 hpA (hours post-amputation) to 26 hpA.

## FACS analysis, isolation of mRNA from macrophages and RT-qPCR

300 *Tg(tnfa:eGFP-F/mpeg1:mCherryF)* larvae were either amputated or infected as described above, then crushed on a 70-µm cell strainer (Falcon, France). Isolated cells were washed in PBS/2 mM ethylenediaminetetraacetic acid (EDTA)/2% Foetal Calf Serum (FCS), filtered through a 40-µm cell strainer, and counted. Counting of mCherry⁺eGFP⁻ and mCherry⁺eGFP⁺ cells was performed on LSRFortessa (BD Bioscience, France), and data analyzed using the Flowjo software (Tree start, Ashland, Or, USA). Sorting was done using FACS ARIA (BD Bioscience, France) and collected in 50% FCS/50% Leibovitz L-15 medium (21083-027, Gibco, France) on ice. To isolate total RNA, cells were

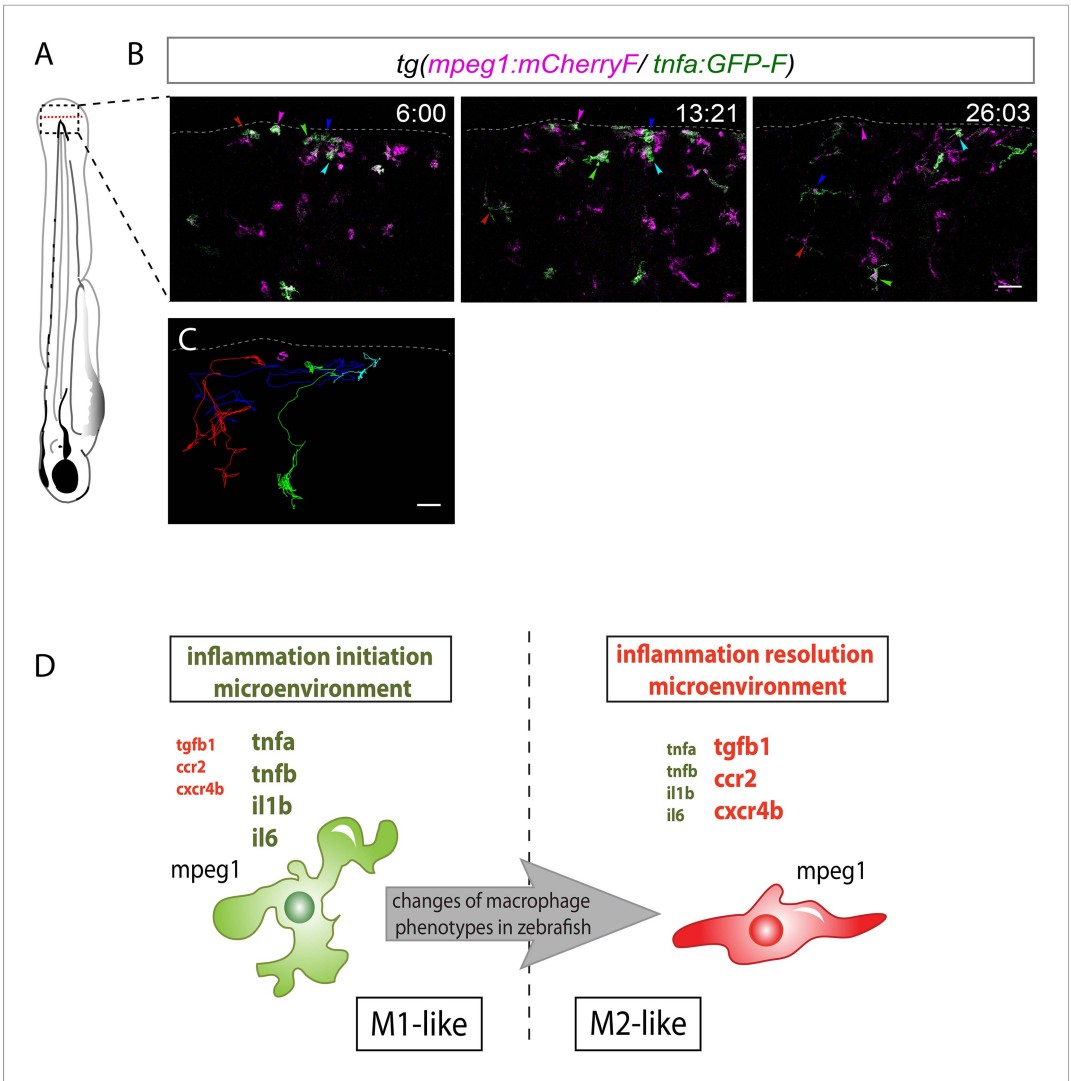

**Figure 4**. M1-like macrophages convert their phenotype toward M2-like phenotype in the wounded fin. (**A**) Diagram showing the site where caudal fin was transected (dotted red line) in 3 dpf *Tg(mpeg1:mCherryF/tnfa:eGFP-F)* larvae. The black dotted box represents the region imaged by confocal microscopy. (**B**) Representative time-lapse maximum projections of 3 dpf *Tg(mpeg1:mCherryF/tnfa:eGFP-F)* amputated fins showing the fate of *tnfa*$^+$ macrophages (magenta + green) at the indicated times pA (hours:minutes) from 6 hpA to 26 hpA. White lines delimit the caudal fin. Scale bar = 30 μm. (**C**) Tracking of *tnfa*$^+$ macrophages from 6 to 26 hpA. The distinct colours of the lines correspond to the distinct macrophages that were indicated with an arrowhead in **B**. (**D**) Diagram representing macrophage activation and polarization in zebrafish. Unpolarized macrophages (*mpeg1*$^+$) are mobilized and recruited to the wound following fin amputation. They are activated and polarized toward a M1-like phenotype (pro-inflammatory) few hours following fin amputation. After 24 hpA, in response to changes in environmental cues, the same macrophages progressively change their phenotype toward intermediate phenotypes and maybe fully polarized M2-like phenotype (non-inflammatory). Main markers of macrophage subtypes are indicated and resemble those found in human (tnfa/b indicates tumour necrosis factor alpha; il1b, interleukin 1-beta; il6, interleukin 6; tgfb1, tumour growth factor beta 1; ccr2, c–c chemokine receptor type 2; cxcr4b, chemokine (C-X-C motif) receptor 4b).

The following figure supplement is available for figure 4:

**Figure supplement 1**. Recruited macrophages remain in the region of tissue injury at 26 hpA.

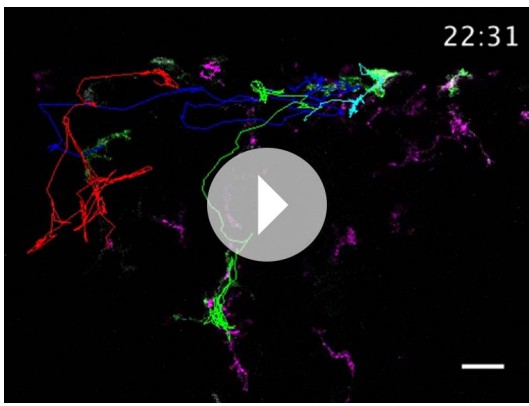

**Video 5.** Recruited GFP⁺ macrophages persist in the region of tissue injury at 26 hpA. *Tg(mpeg1:mCherryF/tnfa:eGFP-F)* caudal fins were amputated at 3 dpf. Representative time-lapse maximum projections show the movements of *tnfa⁺mpeg1⁺* macrophages (green + magenta), starting 6 hpA to 26 hpA. Coloured lines correspond to eGFP⁺ macrophage tracking that stay in the wounded fin and still express eGFP at 26 hpA. Scale bar = 30 μm. Image stacks were acquired every 4 min 40 s at 3.5-μm intervals at 512 × 512 pixel resolution using a confocal microscope TCSSP5 SP5 inverted equipped with a HCXPL APO 40×/1.25–0.75 oil objective (Leica). Excitation wavelengths used were 488 nm for EGFP-F and 570 nm for mCherryF.

lysed in QIAzol Lysis Reagent (Qiagen, France) and RNA extracted using miRNeasy mini kit (Qiagen-21704, France). 20 ng of total RNA was reverse transcribed using High Capacity RNA Reverse Transcription kit (Applied Biosystems, France). QPCR were performed on a LightCycler 480 system (Roche, France), following manufacturer's instructions (SYBR Green format, Roche Applied Science, Meylan, France) and using primers in *Supplementary file 1*: denaturation 15 s at 95°C, annealing 10 s at 64°C, and elongation 20 s at 72°C. Expression levels were determined with the LightCycler analysis software (version-3.5) from 5 independent experiments. The relative amount of a given mRNA was calculated by using the formulae $2^{-\Delta Ct}$ with *ef1a* as reference.

## Statistical analysis

Significance testing for *Figures 1M, 3D–H* was done using Mann–Whitney unpaired t-test, one-tail and *Figure 2F–H* using Mann–Whitney unpaired t-test, two-tails using GraphPad Prism 6 Software. *$p < 0.05$, **$p < 0.01$, ****$p < 0.0001$.

## In situ hybridization

A *tnfa* probe was amplified from total cDNA by PCR using tnfa.55 and tnfa.58 primers (*Supplementary file 1*) and cloned in plasmid pCRII-TOPO. Digoxigenin (DIG)-labelled (Roche, France) sense and anti-sense RNA probes were in vitro transcribed (Biolabs, France). In situ hybridizations on whole-mount embryos were as previously described (*Nguyen-Chi et al., 2012*). For simultaneous detection of eGFP-F proteins and *tnfa* mRNA by immuno-detection and in situ hybridization, fixed and rehydrated *Tg(tnfa:eGFP-F)* larvae were permeabilised in ice in 100% ethanol for 5 min, then in a mixture of 50% Xylene-50% ethanol for 1 hr and in 80% acetone for 10 min at −20°C as described in *Nagaso et al. (2001)*. After washes in PBS-0.1% Tween, larvae were post-fixed in 4% paraformaldehyde (PFA) for 20 min. Subsequent steps of hybridization, washes, and staining with NBT-BCIP (Roche, France) were as previously described in *Nguyen-Chi et al. (2012)*. Next, unspecific-binding sites were saturated in PBS-1% bovin serum albumin (BSA)-1% lamb serum-10% Goat serum and larvae incubated 3 days with an anti-GFP antibody (MBL, 1/500). After extensive washes, larvae were incubated with a goat anti-rabbit antibody. Stained embryos were imaged using a MVX10 Olympus microscope with MVPLAPO 1× objective and XC50 camera and using a Zeiss Axioimager with a Zeiss 40× Plan-Apo 1.3 oil objective.

## Photoconversion of macrophage-specific Kaede protein

*Tg(mpeg1:GAL4/UAS:Kaede)* embryos were raised to 3 dpf in the dark, and caudal fin was transected as described above. At 6 hpA, larvae were mounted in 1% low-melting point agarose. A 405-nm Laser Cube 405-50C on a confocal TCS SP5 inverted microscope with a HCXPL APO 40×/1.25–0.75 oil objective (Leica) was used to photoconvert the Kaede-labelled cells using 6% laser power scanning for 60 s (optimized before the experiments; data not shown). Fins were imaged before and after the photoconversion (at 6 and 26 hpA) in the green and red channels.

## Acknowledgements

This work was supported by Inserm and grants from the Medical Research Foundation (projet FRM 2011 'Comité Languedoc-Roussillon-Rouergue (LRR)', from 'La region Languedoc-Roussillon projet Chercheurs d'avenir 2011' (chercheur avenir 2012-Q-173), from the French National Research Agency for 'Zebraflam' program n° ANR-10-MIDI-009. We thank Myriam Boyer-Clavel from the Montpellier

RIO Imaging platform (MRI) for their help with cytometry and Pr Christine Dambly-Chaudière, U5235/CNRS, France for their help with in situ hybridization.

## Additional information

### Funding

| Funder | Grant reference | Author |
|---|---|---|
| Conseil Régional Languedoc-Roussillon | chercheur avenir 2012-Q-173 | Farida Djouad |
| Agence Nationale de la Recherche | ANR-10-MIDI-009 | Jean-Pierre Levraud, Georges Lutfalla |

Conseil Régional Languedoc-Roussillon: supported Farida Djouad's work by covering the expenses for reagent, imaging and animal facilities as well as the salary for an engineer. Agence Nationale de la Recherche: supported Jean-Pierre Levraud's work by covering animal facility expenses and expenses for reagent and Georges Lutfalla's work by covering animal facility expenses and some expenses for reagents. The funders had no role in study design, data collection and interpretation, or the decision to submit the work for publication.

### Author contributions

MN-C, BL-B, FD, Conception and design, Acquisition of data, Analysis and interpretation of data, Drafting or revising the article; JT, PL-C, GT, QTP, ID-R, J-PL, GL, Acquisition of data, Analysis and interpretation of data; KK, CJ, Conception and design, Drafting or revising the article

### Ethics

Animal experimentation: All animal experiments described in the present study were conducted at the University Montpellier 2 according to European Union guidelines for handling of laboratory animals (http://ec.europa.eu/environment/chemicals/lab_animals/home_en.htm) and were approved by the Direction Sanitaire et Vétérinaire de l'Hérault and Comité d'Ethique pour l'Expérimentation Animale under reference CEEA-LR-13007.

## Additional files

### Supplementary file

• Supplementary file 1. Genes, accession numbers, and sequences of the primers.

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
