## [Decision Letter]

Thank you for sending your work entitled “From the identification of macrophage subsets in zebrafish to the study of their dynamic polarization” for consideration at *eLife*. Your article has been evaluated by Tadatsugu Taniguchi (Senior editor), a Reviewing editor, and three reviewers. The reviewers all found that the study is important but lacks a couple of key experiments, which will be necessary to support the conclusions.

The Reviewing Editor and the reviewers discussed their comments before we reached this decision, and the Reviewing Editor has assembled the following comments to help you prepare a revised submission.

1) The manuscript reports the findings to imply a transition from M1 to M2 phenotype in the same cell. Except for the indirect evidence that cells with dual markers were detected, no attempt was made to directly demonstrate this transition. Perhaps this could be done using time-lapse imaging to show that the GFP+ cells at 6 hpA can be tracked and are still present at 26 hpA. Alternatively, this might be done using a photoconversion approach – there are already existing zebrafish lines where macrophages express a photoconvertible protein. This is a critical piece of data and important since such a demonstration has not been made in in vivo experiments.

2) Figure 1 does not conclusively demonstrate that the *tnfa:eGFP-F* line recapitulates transcriptional activation of *tnfa*, which is critical to interpret all following results using this line. The attempt to show a correlation between *tnfa* mRNA detection and fluorescent reporter expression is insufficient since many genes can be expected to be upregulated in a similar pattern after amputation or during infection. Furthermore, there is no indication that this experiment represents the analysis of more than 1 fish per condition. At a minimum the authors should:

A) Double-stain for *tnfa* and GFP to show consistent overlap at 6hpA.

B) GFP+ cells should be sorted and shown to express enriched levels of *tnfa*. Although QPCR data for *dbl*^*+*^ cells are shown in Figure 3, it is essential to demonstrate that single GFP+ cells are also enriched for *tnfa* transcript. It would also make more sense to include this data in Figure 1.

Minor comments [abridged]:

Figure 1: does not seem to be an unadulterated zoom of the marked area in 1D.

Figure 2: why doesn't mCherry-F appear to exhibit the same membrane localization as eGFP?

Figure 3: should show percentages of each gated population, and single GFP+ cells should be gated.

Figure 3: statistical analysis showing an “increased level of *tgfb*1” in *dbl*^*+*^ macrophages between 6hpA and 26hpA is missing (subsection “Molecular signature of *tnfa*^*+*^ and *tnfa*^*-*^ macrophage populations”).

Figure 4: “initiation/resolution microenvironement” to “initiation/resolution microenvironment”.

Figure 4: The bottom half of this figure is confusing and could be removed without affecting its message.

Title: consider rewording the Title. In its current form, the Title reads more like that of a review article.

Figure 3: Statistical comparisons within the bar graphs in seem to be inconsistent. Are the *dbl*^*+*^ cells at 6 hpA and 26 hpA not significantly different in their expression of *tnfb*, *il1b*, *il6* and *tgfb1*?

Describe the significance of the observations that velocity, contact frequency and duration are different among macrophage phenotypes?

---

## [Author Response]

*1) The manuscript reports the findings to imply a transition from M1 to M2 phenotype in the same cell. Except for the indirect evidence that cells with dual markers were detected, no attempt was made to directly demonstrate this transition. Perhaps this could be done using time-lapse imaging to show that the GFP+ cells at 6 hpA can be tracked and are still present at 26 hpA. Alternatively, this might be done using a photoconversion approach – there are already existing zebrafish lines where macrophages express a photoconvertible protein. This is a critical piece of data and important since such a demonstration has not been made in in vivo experiments*.

We acknowledge that direct demonstration of the transition from M1 to M2 phenotype is missing. As suggested by the reviewers, we generated the *Tg(mpeg1:GAL4 /UAS:Kaede)* larvae to track Kaede^+^ macrophages from 6hpA to 26 hpA. Photoconversion of macrophages recruited at the wound 6hpA demonstrates that most of macrophages stays at the wounded fin 20 hours later. We show that the same macrophages are present at the wound site during inflammation and its resolution suggesting that the same cell might adopt different phenotypes and functions. This result is presented in Figure 4—figure supplement 1. To directly demonstrate that M1-like GFP^+^ macrophages recruited at 6 hpA are still present in situ 26 hpA, we have carefully tracked them using time lapse imaging of wounded *Tg(mpeg1:mCherryF /TNFa:GFP-F)* fins from 6 to 26 hpA, as proposed by the reviewers. Our data demonstrate that initially recruited GFP^+^ macrophages remain at the injury site and still express the GFP. Moreover, we carefully observed macrophage behavior over time post-amputation. Among GFP^+^ macrophages displaying an amyboid phenotype at the wound site 6 hpA, 50% change toward a fibroblastic phenotype from 11 hpA when found at distance of the wound area. These data are described in the paragraph “Molecular signature of *tnfa*^*+*^ and *tnfa*^*-*^ macrophage populations*”* and in Figure 4 and Movie 5. Also, note additional Methods. All together these results show that M1-like GFP^+^ macrophages expressing high levels of *tnfa* first accumulate at the wound 6hpA (Figure 3) and then convert toward a M2-like GFP^+^ macrophage phenotype in situ 26hpA (Figure 3).

*2)*
Figure 1
*does not conclusively demonstrate that the* tnfa:eGFP-F *line recapitulates transcriptional activation of* tnfa*, which is critical to interpret all following results using this line. The attempt to show a correlation between* tnfa *mRNA detection and fluorescent reporter expression is insufficient since many genes can be expected to be upregulated in a similar pattern after amputation or during infection. Furthermore, there is no indication that this experiment represents the analysis of more than 1 fish per condition. At a minimum the authors should*:

*A) Double-stain for* tnfa *and GFP to show consistent overlap at 6hpA*.

*B) GFP+ cells should be sorted and shown to express enriched levels of* tnfa*. Although QPCR data for* dbl^+^
*cells are shown in*
Figure 3*, it is essential to demonstrate that single GFP+ cells are also enriched for* tnfa *transcript. It would also make more sense to include this data in*
Figure 1.

First, we fully agree that quantitative data of *tnfa* mRNA detection and fluorescent reporter expression analysis should have been included in the original manuscript. Experiments were performed on several fishes and we have now provided the n value for each condition in the text (see subsection “In vivo visualisation of macrophage activation and polarization”).

We also acknowledge that a conclusive demonstration that the tnfa:eGFP-F line recapitulates transcriptional activation of tnfa is missing.

A) As reviewers requested, we performed simultaneous detection of *tnfa* mRNA and GFP-F protein in amputated larvae at 6 hpA and have observed a consistent overlap of *tnfa* and GFP-F signal in the fin, showing the direct correlation of GFP-F and *tnfa* transcriptional activation in the fin of the reporter line (Figure 1). The related text is in the subsection “In vivo visualisation of macrophage activation and polarization”. Also, note additional Methods.

B) We acknowledge that it is essential to demonstrate that GFP^+^ cells are enriched for *tnfa* transcript and that result has to be included in Figure 1. We therefore focused on GFP^+^ cells without taking into account the mpeg1^+^ cells since in agreement with the reviewers we believe that it makes more sense to include this data in Figure 1. We FACS-sorted GFP^+^ cells from wounded *Tg(TNFa:eGFP-F)* 6 hpA and performed qPCR analysis for *tnfa* expression, as recommended by the reviewers. We observed a significant increase (43 fold) of *tnfa* mRNA level in GFP^+^ cells as compared to GFP^-^ cells (Figure 1). Thanks to reviewer requests, we can claim that the *Tg(TNFa:eGFP-F)* reporter line recapitulates transcriptional activation of *tnfa*. The corresponding text is in the aforementioned subsection.

*Minor comments [abridged]*:

Figure 1*: does not seem to be an unadulterated zoom of the marked area in 1D*.

We thank the reviewers for drawing to our attention that the zoom of the marked area in 1D is not unadulterated. Figure 1 and Figure 1 represent two different larvae. Therefore we have removed the white box and modified the legend accordingly.

Figure 2*: why doesn't mCherry-F appear to exhibit the same membrane localization as eGFP?*

Post-translational modification of proteins by the addition of a farnesyl group facilitates their membrane association. As mCherry-F and GFP-F are two different proteins, we are not expecting a complete overlapping of their fluorescence in the macrophage. Although they are both mainly targeted to the membrane, they only partially colocalise (Figure 5). Furthermore, we observed accumulation of mCherry-F in structures inside the macrophage while GFP-F does not. Several factors can explain these differences: 1/ they are not expressed at the same level and 2/ their trafficking toward the membrane may occur differently (mCherry-F may be trapped on its way to the membrane). Membrane targeted mCherry pattern is in sharp contrast with that of the NTR fused with an unmodified form of mCherry. When specifically expressed in macrophages NTR-mCherry is localised in the cytoplasm and is outlined by the GFP-F fluorescence (Figure 5).

Author response image 1.GFP-F is expressed in macrophages in *Tg(tnfa:eGFP-F)*.(A) eGFP-F (Green) and mCherry-F (red) fluorescences were analyzed by confocal microscopy amputated *Tg(mpeg1:mCherryF/tnfa:eGFP-F)* fins at 18 hpA. Both farnesylated fluorescent proteins are localised to the membrane of macrophages and mainly colocalise with the exception of mCherry-F accumulation in a structure inside the macrophage. Separated and merged channels are shown. (B) High magnification of the region boxed in A. (C) eGFP-F (Green) and nitroreductase fused with mCherry (NTR-mCherry, red) expressions were analyzed by confocalmicroscopy in amputated *Tg(mpeg1:GAL4 / UAS:NTR-Cherry/ tnfa:eGFP-F)* fins at 6 hpA. The farnesylated form of the eGFP localises to the membrane of macrophages and outlines cytoplasmic NTR-mCherry. Separated and merged channels are shown. (D) High magnification of the region boxed in A.**DOI:**
http://dx.doi.org/10.7554/eLife.07288.017

Figure 3*: statistical analysis showing an “increased level of* tgfb1*” in* dbl^+^
*macrophages between 6hpA and 26hpA is missing (subsection “Molecular signature of* tnfa^+^
*and* tnfa^-^
*macrophage populations”)*.

The statistical analysis showing an “increased level of *tgfb1*” in *dbl*^*+*^ macrophages between 6hpA and 26hpA is missing since we observed an increased tendency and not a significant increase. Cf. below, in response to the comment for Figure 3, and the text (subsection “Molecular signature of *tnfa*^*+*^ and *tnfa*^*-*^ macrophage populations*”*).

Figure 4*: “initiation/resolution microenvironement” to “initiation/resolution microenvironment”*.

This has been corrected in Figure 4.

Figure 4*: The bottom half of this figure is confusing and could be removed without affecting its message*.

We agree with the reviewers that the bottom half of this figure is a little bit far from our data, in order to avoid any confusion we have now removed this part from the Figure 4 (now Figure 4).

*Title: consider rewording the Title. In its current form, the Title reads more like that of a review article*.

We propose as a new title: “Identification of polarized macrophage subsets in zebrafish”.

Figure 3*: Statistical comparisons within the bar graphs in seem to be inconsistent. Are the* dbl^+^
*cells at 6 hpA and 26 hpA not significantly different in their expression of* tnfb*,* il1b*,* il6 *and* tgfb1*?*

We acknowledge that some statistical analysis was missing or inconsistent and we are grateful to the reviewers for highlighting this point. In order to improve the qPCR analysis, we have now included a fifth independent experiment. The statistical significance of differential expression within a 95% confidence interval between bars was improved and indicated in the Figure 3 and in the figure legend. Therefore, *dbl*^*+*^ cells at 6 hpA and 26 hpA are significantly different for the expression level of *tnfa, il1b, il6, ccr2* and *cxcr4*, but not *tnfb* and *tgfb1*, although a tendency toward differential expression level was observed.

*Describe the significance of the observations that velocity*, *contact frequency and duration are different among macrophage phenotypes?*

We observed that macrophage phenotypes (*tnfa*^*-*^ and *tnfa*^*+*^) present different velocity and interaction characteristics. This data show that these macrophages display different behaviours and strongly suggest the existence of different macrophage subsets displaying different functions in our model. This analysis of the cell behaviour is in strong support of the molecular study performed on *tnfa*^*-*^ and *tnfa*^*+*^ macrophages and the conclusion revealing the existence of macrophage subsets in zebrafish. We further describe the significance of this observation in the main text (subsection “In vivo visualisation of macrophage activation and polarization”).